RESEARCH

# Financial costs and personal consequences of research misconduct resulting in retracted publications

**Abstract** The number of retracted scientific articles has been increasing. Most retractions are associated with research misconduct, entailing financial costs to funding sources and damage to the careers of those committing misconduct. We sought to calculate the magnitude of these effects. Data relating to retracted manuscripts and authors found by the Office of Research Integrity (ORI) to have committed misconduct were reviewed from public databases. Attributable costs of retracted manuscripts, and publication output and funding of researchers found to have committed misconduct were determined. We found that papers retracted due to misconduct accounted for approximately $58 million in direct funding by the NIH between 1992 and 2012, less than 1% of the NIH budget over this period. Each of these articles accounted for a mean of $392,582 in direct costs (SD $423,256). Researchers experienced a median 91.8% decrease in publication output and large declines in funding after censure by the ORI.

**ANDREW M STERN, ARTURO CASADEVALL, R GRANT STEEN AND FERRIC C FANG\***

## Introduction

The retraction of a research article can result from honest or dishonest behavior. Honest retractions resulting from scientific error represent a mechanism for correction of the literature (*Casadevall et al., 2014*). Dishonest retractions are triggered by the discovery of data fabrication or falsification, or other types of misconduct. A recent study of the causes for retraction of scientific articles found that the majority of these result from scientific misconduct (*Fang et al., 2012*).

Research misconduct is a breach of ethics that spans a broad range of offenses from deliberate data fabrication or falsification to plagiarism. These types of misconduct apply to both bench science and clinical research. Some studies indicate that hundreds of thousands of patients have been placed at risk of improper medical care due to enrollment in fraudulent studies or the administration of treatment based on fraudulent studies

(*Steen, 2011*). The factors that lead some investigators to commit misconduct are varied and poorly understood. It has been suggested that pressure to 'publish or perish' during times of scarce resources, increased competition and the winner-take-all economics of research may contribute to an atmosphere in which some individuals are tempted to commit misconduct (*Casadevall and Fang, 2012*). Whether the actual frequency of misconduct is increasing is unknown (*Fanelli, 2013; Steen et al., 2013*), but there is increasing interest in this subject given the occurrence of several high profile cases, the creation of websites that track retracted research, such as *Retraction Watch* (*Oransky and Marcus, 2014*), and a concern that misconduct may be having a detrimental effect on the credibility of biomedical research and the reliability of the scientific literature (*Nussenzveig and Zukanovich Funchal, 2008; Trikalinos et al., 2008; Jha, 2012*).

**\*For correspondence:**
fcfang@uw.edu

**Reviewing editor**: Peter Rodgers, eLife, United Kingdom

It is generally accepted that discovery of research misconduct has serious consequences for those who perpetrate it. However, such consequences, as well as other costs of misconduct to society, have not been systematically studied or quantified. In this study we attempted to measure both the damaging effects of research misconduct on the careers of those who perpetrate it and the direct financial costs resulting from the retraction of scientific articles due to research misconduct.

## Results

### NIH funds spent on research misconduct

We first sought to estimate the direct attributable financial cost to the NIH (which is the primary source of public funds for biomedical research in the US) of publications that were retracted due to misconduct. We examined the text of 291 articles originating from the United States and published between 1992 and 2012 that were retracted for research misconduct (*Fang et al., 2012*), and recorded NIH grant numbers cited in footnotes or acknowledgments. Of the articles included in this analysis, 95.9% were retracted due to data falsification or fabrication, with the remainder involving other forms of serious misconduct such as publication without institutional review board (IRB) approval. Articles retracted due to simple plagiarism or duplicate publication were not included. Analyzed articles included both bench research and clinical studies. Using the publicly available NIH ExPORTER database (*National Institutes of Health, 2013*), the award amounts were totaled across all years from 1992 to 2012 and divided by the number of articles linked to the specific grant number in PubMed, thus obtaining an approximate grant cost per article. To calculate the attributable cost of a given article in dollars, we added the costs per article for all grants cited by that article, and then adjusted for inflation. Through this method we were able to identify NIH funding for 149 of 291 retracted articles, with a median attributable cost of $239,381 per retracted article and a mean of $392,582 (*Figure 1A*, *Figure 1—source data 1*). The distribution of attributable cost across this sample is shown in *Figure 1B*; most articles had an attributable cost between $100,000 and $500,000. One outlier was found to have an attributable cost of $3.6 million (*Potti et al., 2006*). Most of the funding for this article came from a large R01 grant that appears to have supported only this single paper. Questions regarding the research by this group first arose in 2007, and it is likely that the subsequent investigations had a negative effect on the productivity of the grant. Of the 149 articles with at least some amount of NIH funding, grants in the ExPORTER database accounted for all cited funding in 43, representing 29% of the whole. Thus this group of 43 articles can be considered a sample for estimating the true direct attributable cost, as all funding is accounted for. The mean attributable cost for this group was $425,072 per article (*Figure 1A*, right column).

We also estimated the funding totals of all NIH grants that in any way contributed to papers retracted for misconduct. To perform this estimate, we added the total grant funding between 1992 and 2012 for grants that were cited at least once by papers retracted during this period, without normalizing to the total number of papers citing these grants. This figure was determined to be $1,668,680,136 in actual dollars, and $2,324,906,182 when adjusted to 2012 dollars to account for inflation.

### Correlation of attributable cost with impact factor

The impact factor is a bibliometric measure of journal rank based on the number of times that articles in that journal are cited by other articles (Thomson Reuters, Philadelphia, PA). Impact factor exhibits a positive correlation with articles retracted for research misconduct or error (*Fang and Casadevall, 2011*; *Fang et al., 2012*). Here we determined that impact factor also correlates with attributable cost (*Figure 1C*), suggesting that the direct cost to the NIH is higher for articles retracted due to research misconduct published in higher impact journals.

### Impact of research misconduct on subsequent research productivity

An additional result of research misconduct is the damaging consequences to the careers and reputations of those found to have committed misconduct. A recent report demonstrated a significant decline in the citation of an author's work after one of his or her papers was retracted (*Lu et al., 2013*). However, the degree to which retraction of an author's paper affects his or her own subsequent research productivity has not been quantified. We attempted to quantify this effect by calculating the number of publications per year for individual senior authors before and after an ORI finding of misconduct. PubMed searches were performed for 44 faculty members over 3- and 6-year intervals before and after being named in an ORI report (*Figure 2A,C*), using the

**A**

|  | All Retracted Papers | NIH-Funded Only |
|---|---|---|
| Total | $58,494,718.60 | $18,278,131.46 |
| Median | $239,381.06 | $361,905.44 |
| Minimum | $7,061.95 | $38,853.65 |
| Maximum | $3,608,713.94 | $1,544,145.88 |
| Mean | $392,582.00 | $425,072.82 |
| Standard Deviation | $423,256.39 | $329,083.42 |
| N | 149 | 43 |

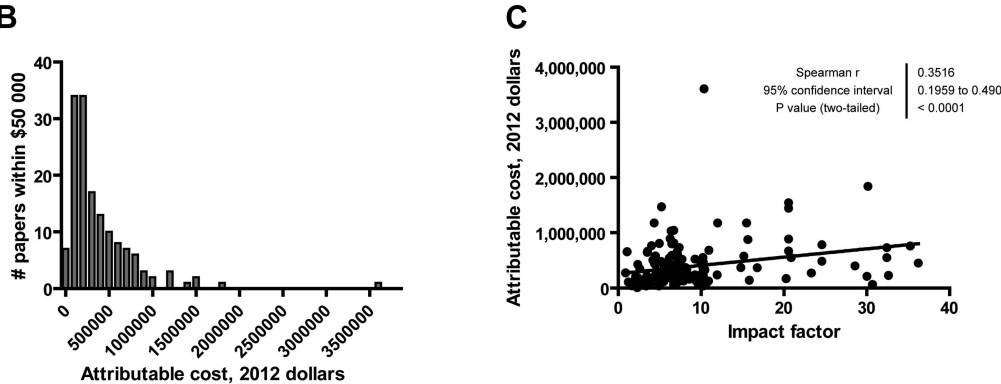

**Figure 1**. Financial costs attributable to research retracted due to misconduct. (**A**) Summary of statistics for articles retracted due to research misconduct between 1992 and 2012. 'NIH-Funded Only' refers to articles that exclusively cited NIH funding sources and for which all supporting grants were retrievable from NIH databases. The complete dataset is available in *Figure 1—source data 1*. (**B**) Histogram depicting the distribution of articles by their individual attributable cost for 149 articles for which at least some NIH funding was cited and retrievable from NIH databases. (**C**) Correlation of attributable cost with impact factor. For articles published during or after 1999, the impact factor for the year of publication was used. For articles published before 1999, the 1999 impact factor was used.

The following source data is available for figure 1:

**Source data 1**. Articles retracted due to research misconduct between 1992 and 2012, and their funding sources.

authors' names and institutions. We found that in most cases, authors experienced a significant fall in productivity following a finding of misconduct. Of the 44 authors with at least one publication in the 3 years before the ORI report, 24 entirely ceased publication in the 3 years after the report, representing 55% of the whole (*Figure 2A*). Similarly, 23 of 44 authors (52%) ceased publication when analyzing 6-year intervals (*Figure 2C*). However, there were several exceptions in which authors continued to publish as much or more than before an ORI finding, suggesting that an instance of misconduct is not necessarily a career-ending event. In addition to a dramatic percent change in publication rate, we also observed a substantial decrease in the total and median number of publications before and after an ORI misconduct finding (*Figure 2B,D*). A total of 256 publications by 54 authors were identified in the 3-year pre-ORI period (median 1.0 per year), but only 78 in the 3-year post-ORI period (median 0 per year), a 69.5% decline (*Figure 2B*). The additional

ten authors in *Figure 2B* had zero publications in the pre-ORI interval and thus no percentage change was calculated for *Figure 2A*. Similarly, there were a total of 552 publications for 47 authors in the 6-year pre-ORI period (median 1.2 per year), but only 140 in the 6-year post-ORI period (median 0 per year), a 74.6% decline (*Figure 2D*). As above, the additional three authors in *Figure 2D* had zero publications in the pre-ORI interval, and thus no percentage change was calculated for *Figure 2C*.

As the identification of an author by institution in a PubMed search might be misleading if researchers are dismissed by an institution after a finding of misconduct, we also used the Author Search from Web of Knowledge (*Thomson Reuters, 2013*), which uses several factors other than institutional affiliation to separate authors with similar names (*Figure 2E*). Authors were retrieved from this search strategy based upon their affiliation with a research field similar to that of a faculty member with the same name who was named in

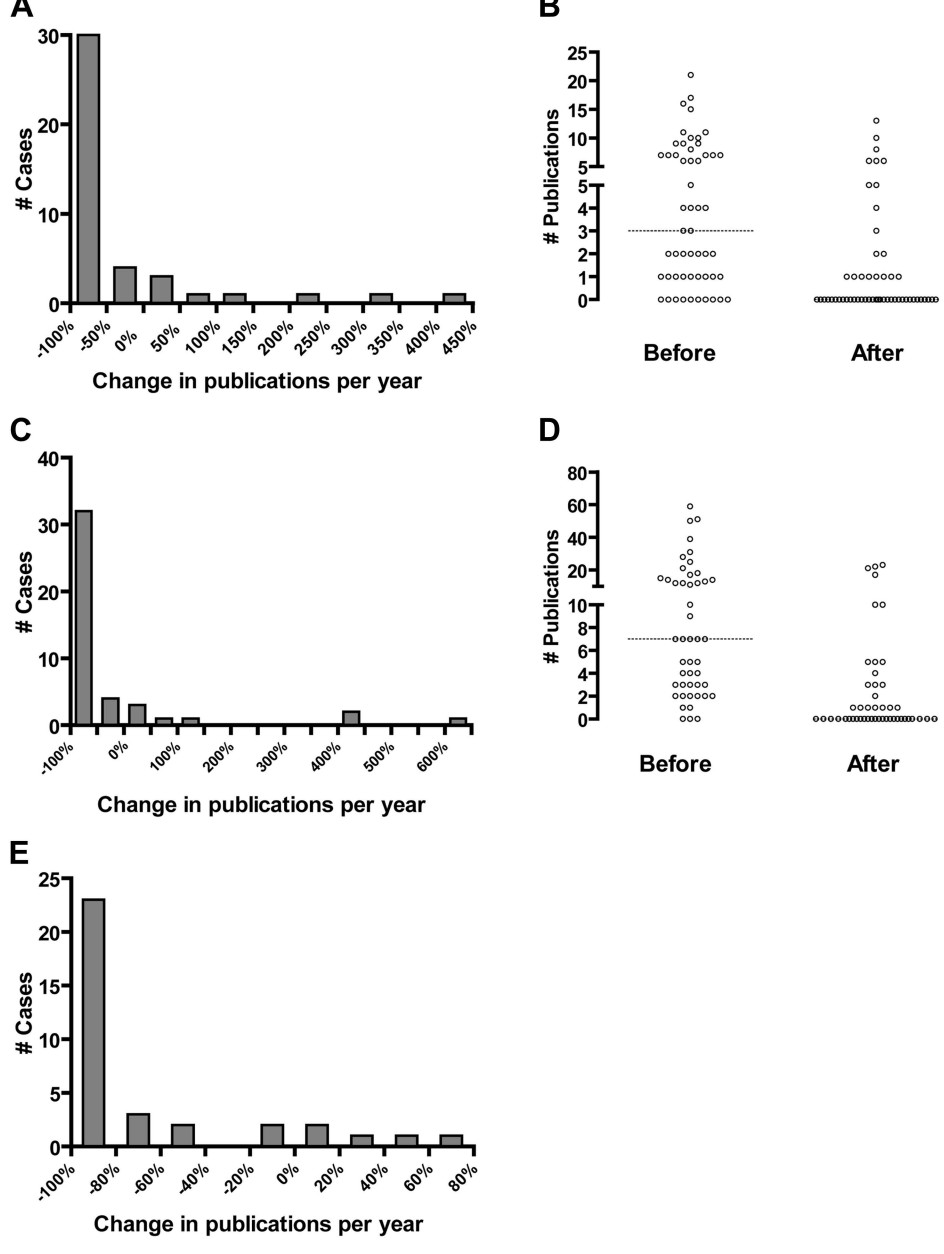

**Figure 2**. Effect of Office of Research Integrity misconduct findings on research productivity. The productivity of principal investigators found to have committed misconduct by the ORI was evaluated by a PubMed search by author name and institution for 3-year (**A** and **B**) and 6-year (**C** and **D**) intervals prior to and following the release of the ORI report, excluding the actual year of the ORI report. Represented are authors with at least one publication in the 3- or 6-year intervals before the ORI report which in both cases totaled 44. (**A** and **C**) Percent change in publications following the ORI report. Most of these authors experienced a large negative change, although some experienced a positive change, primarily those who did not falsify or fabricate data. (**B** and **D**) Absolute number of publications during 3-year (**B**) and 6-year (**D**) intervals before and after the ORI report. Each dot represents a single investigator before and after the ORI report. Dotted line indicates the median before the ORI report; in both cases the median was zero after the ORI report. (**E**) Productivity of PIs before and after ORI findings of misconduct was analyzed using the Web of Knowledge Author Search. This includes all publications by that author before the ORI finding compared to the interval between the ORI finding and 2012, excluding the actual year of the ORI report.

an ORI finding. In this analysis, all years in which an author was active were analyzed from the year of first publication until 2012. We were able to identify 35 investigators using this method, with a median of 2.9 publications per year before the ORI report and a median of 0.25 publications per

year following the ORI report, representing a median percent change of −91.8%. These results are qualitatively similar to those obtained with PubMed. Collectively these observations suggest that a finding of misconduct by the ORI typically, though not always, results in a severe decline in research productivity.

### Impact of research misconduct on subsequent funding

Censure of a scientist by the ORI for serious infractions usually results in an agreement banning contracts between that scientist and the Public Health Service for a period of time, the length of which is determined by the severity of the infraction. Although it is frequently assumed that ORI citations result in a meaningful decrease in funding as a result of these policies, this has never been quantified. We thus searched the ExPORTER database for NIH grants attributed to PIs found to have committed misconduct by the ORI between 1992 and 2012 (*Figure 3—source data 1*). We then focused on funding during the 5 year intervals before and after the ORI report was published, for ORI reports between 1997 and 2007. Through this method we identified $23,206,662 in funding occurring during the 5 year intervals before ORI reports and $6,842,346 in funding given to the same PIs after the ORI reports were published, a 70.5% decline. The median funding per year per principal investigator (PI) with respect to the ORI report is shown in *Figure 3A*. Not surprisingly, the publication of an ORI report was correlated with a significant and sustained drop in funding. Interestingly, both the total and median funding appeared to decline even before the year of ORI report publication (*Figure 3A,B*). We hypothesize that this may be due to a decline in productivity and funding success during the time in which internal university investigations occur, prior to consultation by the ORI. Alternatively it may be that fiscal stress is a risk factor for PI misconduct.

## Discussion

### Research misconduct accounts for a small percentage of total funding

Increasing concern about research misconduct has raised questions with regard to its impact on the scientific and medical communities, including financial costs. We attempted to determine empirically the amount of money spent by the NIH to support articles that were subsequently retracted due to research misconduct. No such figure has been previously reported, likely because it is

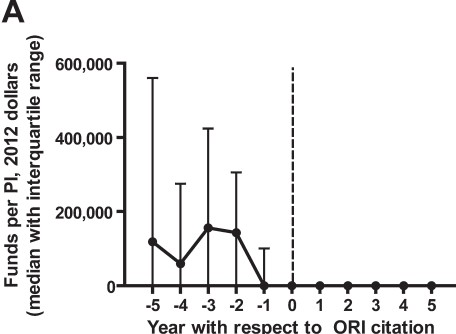

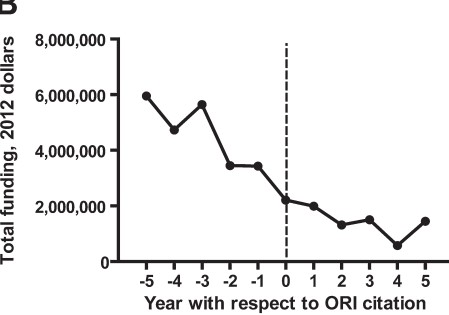

**Figure 3**. Effect of Office of Research Integrity misconduct findings on funding. The ExPORTER database was searched for PIs found to have committed misconduct by the ORI, and their funding totals by year were aligned with respect to the year of citation by the ORI. This was performed for ORI reports published between 1997 and 2007. Shown are median (**A**) and total (**B**) funding by the NIH to PIs found by the ORI to have committed misconduct, with respect to the year of the ORI report. The complete dataset is available in *Figure 3—source data 1*.

The following source data is available for figure 3:

**Source data 1**. PIs found by the ORI to have committed misconduct between 1992 and 2012, and their NIH funding support.

difficult to calculate the attributable cost of a particular study. Funding sources are diverse and are not universally reported in manuscripts or in public databases. Multiple grants may be used to fund many different studies, so it is difficult to specifically determine the funds used to support a given article. We focused on research supported by the NIH, which publishes ExPORTER, a publicly accessible database of all NIH-supported grants. The $46.9 million in funds (not inflation-adjusted) used to support 149 articles retracted due to research misconduct represents only about 0.01% of the total $451.8 billion NIH budget between 1992 and 2012.

To verify that bias in our study did not lead to significant underestimation of the true financial cost of research misconduct, we attempted to

identify several reasons why our calculations might be too low. First, many studies retracted due to misconduct did not state their funding sources, or may have stated them incompletely. We attempted to mitigate this effect by re-calculating research costs for the 43 articles that exclusively cited NIH funding sources found in ExPORTER. When considering only these articles, the mean attributable cost per retracted article increased from $392,582 to $425,073. Multiplying this number by the total number of articles retracted due to scientific misconduct in the United States (291) yields a value of $123.7 million, which might be considered an estimate of the total NIH funds directly spent on known biomedical research retracted due to misconduct over the past 20 years. Additionally, even when deliberately overestimating the funding contributions to this research by totaling all grants that provided any support for papers retracted due to misconduct, the figure ($1.67 billion in actual funds, $2.32 billion in 2012 dollars) is still less than 1% of the NIH budget over this time period.

Second, our analysis only accounted for research misconduct that was detected and investigated. If retracted papers represent only the 'tip of the iceberg' of a much greater amount of misconduct that occurs nationally, then our analysis would miss this significant portion. A meta-analysis of survey data indicates that approximately 2% of scientists admit to data falsification or manipulation on at least one occasion (*Fanelli, 2009*), whereas less than 0.02% of published data are retracted due to fabrication, falsification, or plagiarism (*Claxton, 2005*; *Fang et al., 2012*). Thus, if only 1% of research misconduct is actually detected, then the estimated amount of funding dedicated to work resulting from misconduct would extrapolate to approximately $12.4 billion based on our estimate of $123.7 million. Even with this correction, research misconduct would only account for approximately 1.5% of the NIH research budget since 1992. In our view, this is still a relatively low number, suggesting that research misconduct does not involve a large percentage of research funding in the United States.

In fact, our analysis could represent an overestimation rather than underestimation of the true cost of scientific misconduct. We make the assumption that every dollar spent on a publication retracted due to scientific misconduct is 'wasted'. However, it is conceivable that some of the research resulting in a retracted article still provides useful information for other non-retracted studies, and that grant funds are not necessarily evenly distributed among projects and articles. Moreover, laboratory operational costs for a retracted paper may overlap with those of work not involving misconduct. Thus, considering every dollar spent on retracted publications to be completely wasted may result in an overestimation of the true cost of misconduct.

Some additional sources of bias in our financial analysis should be acknowledged. Funding data obtained from ExPORTER from 1992–2012 were used to analyze articles over the same interval. However, grants from recent years have funded some articles that have not yet been published. Thus, the denominator for calculations of attributable cost for these grants may be artifactually low, inflating the calculated attributable costs for recent articles funded by these grants. This might be offset by articles from the beginning of the interval, which would have a smaller numerator, since grant years before 1992 were not included.

An interesting question generated by our analysis is whether papers retracted due to misconduct cost, on average, more or less than other papers. This might indicate whether misconduct accounts for a disproportionate percentage of funds. On one hand, retracted papers might require fewer materials or personnel if the data are simply fabricated, and thus cost less to publish. On the other hand, papers retracted due to misconduct might be generated by authors who have spent large amounts of funding on otherwise unsuccessful experiments, and thus account for a large percentage of their grants. This question is unanswered at the present time.

### Research misconduct entails other financial costs

The financial costs of research misconduct extend beyond the grants supporting the work. Investigations of misconduct allegations are costly for institutions. In analyzing a single case at the Roswell Park Cancer Institute, *Michalek et al. (2010)* identified many financial costs of research misconduct, including legal and consulting fees, salaries for faculty and witnesses involved in the investigation, institutional review board costs and other expenses, with estimates exceeding $500,000, greater than the median attributable cost to research funding sources. These authors also identified other indirect costs, such as damage to institutional and lab members' reputations that might affect future funding. *Gammon and Franzini (2013)* analyzed 17 cases using a mixed methods model to conclude that ORI cases incur financial costs ranging from

approximately $100,000 to $2 million. These authors analyzed the direct NIH funding costs as well as the cost of the ORI investigation itself, costs to the institution for monitoring the work of the faculty after ORI censure, and the cost of retracting articles.

One must also consider the cost of unproductive research by other scientists who have based their work on retracted publications. This indirect financial cost due to the reverberations of fraud throughout the research community might be even greater than the cost of the retracted research itself. We did not measure these costs in our analysis, which was designed as an empiric measurement of financial costs using actual funding totals. However, although other indirect costs cannot be measured directly and were not included in this study, they may nevertheless account for a large additional cost of research misconduct.

### Higher financial costs are associated with higher-impact journals

We observed that the attributable costs of retracted manuscripts correlated with the impact factors of the journals in which they were published. There are multiple possible explanations for this observation. One is that high-impact publications tend to require more data for acceptance, which would in turn increase the amount of funds devoted to that study. Another possibility is that authors would be more likely to list a grant on a high-impact publication in order to promote the success of the grant in future applications for renewal. These explanations could apply to all manuscripts, regardless of whether or not they were retracted. Nevertheless, our data demonstrate that future retracted publications from high-impact journals are likely to have required more funding, on average, than retracted publications from low-impact journals.

### Research misconduct correlates with decreased productivity and funding

The personal consequences for individuals found to have committed research misconduct are considerable. When a researcher is found by the ORI to have committed misconduct, the outcome typically involves a voluntary agreement in which the scientist agrees not to contract with the United States government for a period of time ranging from a few years to, in rare cases, a lifetime. Recent studies of faculty and postdoctoral fellows indicate that research productivity declines after censure by the ORI, sometimes to zero, but that many of those who commit misconduct are able to find new jobs within academia (*Redman and Merz, 2008*, *2013*). Our study has found similar results. Censure by the ORI usually results in a severe decrease in productivity, in many cases causing a permanent cessation of publication. However the exceptions are instructive. Of 35 faculty ORI cases analyzed using Web of Knowledge, five actually published more articles per year after an ORI report than before: Raphael Stricker (*ORI, 1993*), Gerald Leisman (*ORI, 1994*), Oscar Rosales (*ORI, 1995*), Ruth Lupu (*ORI, 1996*), and Alan Landay (*ORI, 1995*). Of these, only Stricker was found to have falsified or fabricated data; the other four were found to have falsified letters of collaboration or to have committed plagiarism, which might be considered lesser infractions. (Even though Stricker left his academic position following the finding of misconduct, he continued to publish actively, although more than half of these publications were correspondence or commentaries).

Scientists who falsified or fabricated data generally experienced severe drop-offs in productivity. Our results suggest that a finding of misconduct by the ORI significantly reduces research productivity. We did not examine the important additional effect of collateral damage to other researchers associated with an investigator found to have committed misconduct, but anecdotal reports suggest that these consequences can also be substantial and even career-threatening (*Couzin, 2006*; *Nature, 2010*). Our study documents that censure by the ORI results in a significant decline in both citation productivity (*Figure 2*) and the ability to acquire NIH funding (*Figure 3*).

## Conclusions

This study provides an analysis of two important effects associated with research misconduct resulting in the retraction of scientific publications: financial costs to funding agencies and damage to the research careers of those who commit misconduct. We found that research misconduct indeed incurs significant financial costs, although the direct financial costs to the NIH were modest when viewed as a fraction of the total research budget. The greatest costs of misconduct, preventable illness or the loss of human life due to misinformation in the medical literature, were not directly assessed. For example, a now-discredited article purporting a link between vaccination and autism (*Wakefield et al., 1998*) helped to dissuade many parents from obtaining vaccination for their children. Decreasing vaccination rates are often associated

with outbreaks of preventable infections, such as a recent measles outbreak in Wales that resulted in more than 1200 cases and cost an estimated £470,000 (~$800,000 US) (*McWatt, 2013*). In another high-profile fraud case involving studies of the use of colloid for fluid resuscitation, there is evidence that the discredited body of work led to the use of a therapy with higher mortality and morbidity than standard care (*Wise, 2013*; *Zarychanski et al., 2013*). Thus, although we estimate that only a very small percentage of NIH grant dollars has been spent on research misconduct, the indirect costs to society are likely to be substantially greater. Further empirical study of research misconduct may help to more fully elucidate its serious consequences, whether it is truly increasing, and how it can be prevented.

## Materials and methods

### Inclusion criteria

A comprehensive search for retracted articles was previously performed and filtered specifically for articles retracted due to documented or suspected serious research misconduct (*Fang et al., 2012*), primarily consisting of data fabrication or falsification (95.9%). Articles retracted due to simple plagiarism or duplicate publication were not included. We focused on papers from the United States because this country has an Office of Research Integrity with publicly-accessible records that allowed this study. We selected the 290 articles from this database that were retracted due to misconduct originating from the United States between 1992 and 2012, as well as one additional article originating from France (*Le Page et al., 2000*) that was retracted due to data falsification by a co-author (Steven Leadon) working in the United States.

### Calculation of attributable costs

The text of each article was examined and funding sources identified by the authors were noted. Any cited NIH grant numbers were used to search the ExPORTER database (*National Institutes of Health, 2013*) for award totals from 1992 to 2012, and all years were totaled to obtain the total grant cost over this interval. A PubMed search was then performed using the standard format:

XX0##### OR XX##### AND('1992'[Date–Publication]: '2013'[Date–Publication]) NOT 'Retraction'(Title).

where 'XX' refers to the institute code (for example, 'CA' for the National Cancer Institute) and

'#####' refers to the grant number. Although a full grant number contains six digits, most grant numbers are cited as five digits with omission of the initial 0, so both possible terms were included. For recent grant number beginning with 1, the terms 'XX1##### OR XX1#####' were used instead. In cases in which a grant's fiscal year was specified by the authors (for example, XX#####-01), the entire grant was used due to difficulties performing PubMed searches for specific fiscal years. To calculate the attributable cost per article for an individual grant, the total amount of the grant from the ExPORTER database search was divided by the number of articles obtained from the PubMed search. The attributable cost for a given article was calculated as the sum of the attributable costs per article of the NIH grants cited by the article. The resulting figures were then adjusted using the US Bureau of Labor Statistics inflation calculator (http://www.bls.gov/data/inflation_calculator.htm) by inputting the year of publication and adjusting to 2012 dollars.

### Enumeration of publications

Two methods were used to determine the number of times that an author has published, in order to limit searches to publications written by a single author with a common name. First, PubMed was searched by institution:

(Lastname IN[Author]) AND institution(Affiliation) AND ('XXXX'[Date–Publication]: 'YYYY'[Date–Publication]).

where 'IN' refers to the author's initials, and 'institution' refers to keyword(s) specifying the author's institution. XXXX and YYYY are years specifying intervals before and after the year of an ORI report, as indicated in the figure legends. Second, since authors may change institutions, the Web of Knowledge (*Thomson Reuters, 2013*) 'Author Search' tool was also applied. This tool uses several factors to distinguish individual authors with similar names. An author's last name and initials retrieve citations for individuals in a research field corresponding to other articles known to be published by that author.

Despite these precautions, some articles may have been incorrectly attributed to authors, and some articles authored by an author may have been missed. This problem has been noted previously (*Falagas, 2006*; *Bourne and Fink, 2008*; *Lane, 2010*; *Polychronakos, 2012*), and systems have been proposed for the disambiguation of authors with similar names (*Fenner, 2011*). However, none of these has been universally adopted.

## Calculation of funding with respect to year of ORI report

To determine the effect of ORI citation on a principal investigator (PI)'s funding, we first searched the ExPORTER database for PIs who were cited for research misconduct by the ORI, and recorded the total funding for all grants to that PI between 1992 and 2012. In the case that a PI changed institutional affiliations during this period, we checked the titles of all grants to ensure they were either the same grant carried to a new institution or, if new grants, were within the same area of expertise. Grants totals were then converted to 2012 dollars using the Bureau of Labor Statistics inflation calculator. We then aligned the yearly funding totals for all PIs with respect to the year of the ORI report. To avoid bias due to ORI reports occurring close to the limits of our 20-year time window, we focused on the 5-year window before and after the year of the ORI report only for ORI reports between 1997 and 2007, and calculated median and total funding during these intervals. In calculating the median funding shown in *Figure 3A*, we included PIs who had no funding between 1997 and 2007 but had at least some funding in the ExPORTER database between 1992 and 2012.

## Acknowledgements
We thank John Ioannidis for valuable insights into this manuscript.

**Andrew M Stern** Perelman School of Medicine, University of Pennsylvania, Philadelphia, United States
**Arturo Casadevall** Department of Microbiology and Immunology, Albert Einstein College of Medicine, New York, United States; Department of Medicine, Albert Einstein College of Medicine, New York, United States
**R Grant Steen** MediCC!, Medical Communications Consultants, LLC, Chapel Hill, United States
**Ferric C Fang** Department of Laboratory Medicine, University of Washington, Seattle, United States; Department of Microbiology, University of Washington, Seattle, United States

## Author contributions
AMS, AC, FCF, Conception and design, Acquisition of data, Analysis and interpretation of data, Drafting or revising the article; RGS, Analysis and interpretation of data, Drafting or revising the article

*Competing interests:* The authors declare that no competing interests exist.

## Funding

| Funder | Author |
| --- | --- |

No external funding was received.

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
