## [Decision Letter]

Thank you for sending your work entitled “Personal and Financial Costs of Research Misconduct Resulting in Retracted Publications” to *eLife* for consideration as a Feature Article. Your article has been favorably evaluated by the Features Editor (Peter Rodgers) and three outside reviewers, two of whom, Daniele Fanelli and Xavier Bosch, have agreed to reveal their identity.

The three reviewers and the Features Editor discussed their comments before reaching this decision, and the Features Editor has assembled the following comments to help you prepare a revised submission.

This is a well-written paper, which presents an interesting new study attempting to estimate some of the costs of scientific misconduct. In particular, the authors aim to evaluate how much the research funds are wasted due to fraudulent research, and then add what they define as “personal costs” in terms of reduced productivity and grant funding access for those found guilty of misconduct. The authors elaborate and discuss in detail the limitations and potential biases and their conclusions are coherent with the main results.

However, the reviewers were concerned about the following points.

It is essential that you address the following comments (3; 11; 33; 2; 9; 32) in the revised version of the manuscript.

1) Calculating the financial costs

# The authors should clarify what is included in grant costs – indirect and direct, or just direct. If the cost is just direct, why are indirect costs excluded?

# Also, the change in the value of the dollar over the period of the study affects the conclusions regarding ”cost“. Tom's Inflation Calculator can ”correct“ dollars so the authors are comparing apples to apples. These grant costs must be adjusted for the long study period.

# Ideally, the paper should partition the analyses by type of misconduct, attempting to control for confounding effects. If this is too challenging, it should discuss possible limitations cause by the lack of this disambiguation.

# The case becomes ambiguous when retractions are due to forms of misconduct other than fabrication and falsification. Please discuss the following questions: Does a paper that is retracted because of duplicate publication or plagiarism cost anything at all in terms of research funds and research results? And what about retractions due to breaches of ethical standards?

# In the “Impact of research misconduct on subsequent research productivity”, did authors examine whether the PI status of PhD or MD was a variable in the continued publication after the ORI finding? [Note: You do not need to include such an analysis in the revised version.] Did literature searches include any category of authorship? Or, were first author only searches conducted?

2) Terminology

“retracted due to misconduct”: Judging by the data file, the authors seem to use the term misconduct as synonymous with data fabrication and falsification – i.e., different from plagiarism. This breaks a technical convention and might create confusion in the reader. Clarity should be made in the text by using the technical terms (fabrication and falsification) consistently when appropriate, and term misconduct to indicate the entire class of fraudulent behaviours (FFP).

3) Previous work that should be cited

# The claim that misconduct may not have been growing is given no supporting reference. Some of the authors have, in the past, repeatedly suggested the opposite. I (DF) have been involved in an open academic dispute with these authors around this point, and have published a recent paper giving various lines of support to the idea that we cannot tell if misconduct is growing or not (D Fanelli 2013, Why Growing Retractions Are (Mostly) a Good Sign, PLOS Medicine doi: 10.1371/journal.pmed.1001563).

# You introduced the work of Michalek et al. to consider measureable financial costs of research misconduct in a single case. Gammon and Franzini identified grant costs in Research Misconduct Oversight: Defining Case Costs, J Health Care Finance 2013; 40(2):75-99. They use a sequential mixed method study design to define economic costs of research misconduct in 17 cases of misconduct and include grant costs.

# Please consider if any of the following papers should also be cited:

Glick JL. 1989. On the Potential Cost Effectiveness of Scientific Audit. Accountability in Research 1:77-83.

Glick JL, Shamoo AE. 1991. Auditing Biochemical Research Data: A Case Study. Accountability in Research 1:223-243.

Shamoo AE. 2013. Data audit as a way to prevent/contain misconduct. Accountability in Research 20:369-79

Shamoo AE. 2006. Data audit would reduce unethical behaviour. Nature 439:784

4) Correlation with impact factor

There are many reasons why the rate of retractions might scale with journal impact factor, and many reasons why articles in journals with high impact factors might cost more than those in journals with low impact factors (irrespective of whether or not they have been retracted). Please either discuss other possible explanations of Figure 1 in the text, or remove Figure 1 and the related parts of the text.

5) Personal costs

A major concern relates to the fact that the study deals with two fully different issues: financial costs and personal costs, which could indeed make two papers. The strength of the paper (subject to some comments above) is the economic repercussions of retractions from a misconduct finding. However, the term “personal costs” as defined and analyzed by the authors is misleading: it is not the wasted money of US taxpayers, it is the consequences one would expect to follow for someone who has engaged in misconduct when their fraud is uncovered and made public by a government agency. Perhaps “personal consequences” would be a more appropriate expression.

6) Discussion

# “The $46.9 million in funds used to support 149 fraudulent articles represents only about 0.01% of the total $451.8 billion NIH budget between 1992 and 2012”: as the authors point out later on, this percentage doesn't mean much, since it would in all cases reflect the (minuscule) percentage of papers that are actually retracted. It would be interesting to see if the proportion of funding wasted is higher than the relative proportion of papers funded by NIH that are retracted. This might be a challenging calculation to perform, and maybe the topic of another study, but I would encourage the authors to discuss the point.

# “As our estimates of attributable costs are a small percentage of the NIH budget, we identify several reasons why our calculation may underestimate the true financial cost of research misconduct”: the authors do, overall, a thorough job in discussing their results, and show academic rigour in reaching a conclusion that was presumably against their initial hypothesis. Within this context, I find the tone of this sentence a bit out of place. It suggests that the authors, disappointed with the modest costs they had calculated, went on to seek any reason to believe that they underestimated the true costs. In reality, there are reasons (which I point out above) why the costs could have been overestimated as well. Upon revision, more reasons why costs could have been over-estimated could be discussed.

“If only 1% of research misconduct is actually detected … suggesting that research misconduct does not involve a large percentage of research funding in the United States”: This passage is somewhat laboured and would benefit from being shortened.

“…$1.7 billion in fraudulent expenditures were recovered by Medicaid fraud control units…”: The authors' inclusion, exclusion logic is not apparent. Why not include Medicare fraud? Why mix military misappropriations with research fraud? We suggest deleting this passage.

“This indirect financial cost due to the reverberations of fraud throughout the research community is likely to be substantially greater than the cost of the fraudulent research itself”: too speculative, not supported by evidence. I agree that indirect costs of say grants awarded to other scientists based on papers retracted due to misconduct by primary offenders might be high. But since this has not been analyzed, it may be more prudent to say “….throughout the research community 'might be even greater' and not 'is likely to be substantially greater' than the cost…”

---

## [Author Response]

1) Calculating the financial costs

# The authors should clarify what is included in grant costs – indirect and direct, or just direct. If the cost is just direct, why are indirect costs excluded?

We included only direct costs in this study, as reported by the NIH in their ExPORTER database. Direct costs are measured directly, whereas indirect costs can only be estimated, as studies have shown that indirect grant revenue underestimates the true indirect costs of research (Dorsey et al., Acad Med, 2009). Inclusion of indirect costs would therefore change our study from an empiric one to an estimation model. This is emphasized in the Discussion section in the revised manuscript.

*# Also, the change in the value of the dollar over the period of the study affects the conclusions regarding “cost”. Tom's Inflation Calculator can “correct” dollars so the authors are comparing apples to apples. These grant costs must be adjusted for the long study period*.

We have now corrected the data in Figures 1 and 3 for inflation by converting all values to 2012 dollars. This is reflected in the text of the revised manuscript.

*# Ideally, the paper should partition the analyses by type of misconduct, attempting to control for confounding effects. If this is too challenging, it should discuss possible limitations cause by the lack of this disambiguation*.

Only articles retracted due to documented or suspected serious misconduct were included in this study; 95.9% of the articles were retracted due to data falsification or fabrication. Articles retracted for simple plagiarism or duplicate publication were not included. Therefore it is not possible to partition the analysis by type of misconduct.

# The case becomes ambiguous when retractions are due to forms of misconduct other than fabrication and falsification. Please discuss the following questions: Does a paper that is retracted because of duplicate publication or plagiarism cost anything at all in terms of research funds and research results? And what about retractions due to breaches of ethical standards?

We agree that the relative cost to funding sources may differ depending on the type of misconduct but we do not have data to address this question. Most (95.9%) of the articles in this study were retracted due to documented or suspected data falsification or fabrication. We did not analyze the costs of retractions resulting from other types of misconduct, such as simple plagiarism or duplicate publication. However we suggest that any dollar spent on a study retracted due to misconduct in any form represents a waste of resources.

# In the “Impact of research misconduct on subsequent research productivity”, did authors examine whether the PI status of PhD or MD was a variable in the continued publication after the ORI finding? [Note: You do not need to include such an analysis in the revised version.] Did literature searches include any category of authorship? Or, were first author only searches conducted?

We did not analyze the relationship between academic rank or degree and continued publication. Literature searches were not restricted by category of authorship, but were restricted to the individual faculty author found to be responsible for misconduct.

2) Terminology

*“retracted due to misconduct”: Judging by the data file, the authors seem to use the term misconduct as synonymous with data fabrication and falsification - i.e. different from plagiarism. This breaks a technical convention and might create confusion in the reader. Clarity should be made in the text by using the technical terms (fabrication and falsification) consistently when appropriate, and term misconduct to indicate the entire class of fraudulent behaviours (FFP)*.

Text has been added to the Methods and Results sections further explaining what was included in our sample. We made changes throughout the text to avoid use of the term “fraud,” which may have a vague meaning to some readers.

3) Previous work that should be cited

*# The claim that misconduct may not have been growing is given no supporting reference. Some of the authors have, in the past, repeatedly suggested the opposite. I (DF) have been involved in an open academic dispute with these authors around this point, and have published a recent paper giving various lines of support to the idea that we cannot tell if misconduct is growing or not (D Fanelli 2013, Why Growing Retractions Are (Mostly) a Good Sign, PLOS Medicine* doi: 10.1371/journal.pmed.1001563*)*.

We now cite two articles in this sentence (the one pointed out by Prof. Fanelli as well as an article by Steen et al.), which represent divergent viewpoints on this issue. It is our view that current data cannot distinguish between these possibilities, and it is likely that both increasing rates of misconduct and increasing detection are contributing factors to the observed trend.

*# You introduced the work of Michalek et al. to consider measureable financial costs of research misconduct in a single case. Gammon and Franzini identified grant costs in Research Misconduct Oversight: Defining Case Costs, J Health Care Finance 2013; 40(2):75-99. They use a sequential mixed method study design to define economic costs of research misconduct in 17 cases of misconduct and include grant costs*.

We thank the reviewers for bringing our attention to this interesting study. We have added it to the cited references.

# Please consider if any of the following papers should also be cited:

*Glick JL. 1989. On the Potential Cost Effectiveness of Scientific Audit. Accountability in Research 1:77-83*.

*Glick JL, Shamoo AE. 1991. Auditing Biochemical Research Data: A Case Study. Accountability in Research 1:223-243*.

Shamoo AE. 2013. Data audit as a way to prevent/contain misconduct. Accountability in Research 20:369-79

Shamoo AE. 2006. Data audit would reduce unethical behaviour. Nature 439:784

The above papers discuss independent audits of scientific research as one of many potential solutions to the problem of research misconduct. As our manuscript attempts to quantify one aspect of this problem but does not discuss solutions, we would prefer not to cite these papers.

4) Correlation with impact factor

*There are many reasons why the rate of retractions might scale with journal impact factor, and many reasons why articles in journals with high impact factors might cost more than those in journals with low impact factors (irrespective of whether or not they have been retracted). Please either discuss other possible explanations of*
Figure 1
*in the text, or remove*
Figure 1
*and the related parts of the text*.

We would prefer to retain this figure in the manuscript. Possible explanations for this correlation have been added to the Discussion section.

5) Personal costs

*A major concern relates to the fact that the study deals with two fully different issues: financial costs and personal costs, which could indeed make two papers. The strength of the paper (subject to some comments above) is the economic repercussions of retractions from a misconduct finding. However, the term “personal costs” as defined and analyzed by the authors is misleading: it is not the wasted money of US taxpayers, it is the consequences one would expect to follow for someone who has engaged in misconduct when their fraud is uncovered and made public by a government agency. Perhaps “personal consequences” would be a more appropriate expression*.

We have changed the terminology to “personal consequences” or “damage to careers” throughout the manuscript, as the reviewer suggests.

6) Discussion

*# “The $46.9 million in funds used to support 149 fraudulent articles represents only about 0.01% of the total $451.8 billion NIH budget between 1992 and 2012”: as the authors point out later on, this percentage doesn't mean much, since it would in all cases reflect the (minuscule) percentage of papers that are actually retracted. It would be interesting to see if the proportion of funding wasted is higher than the relative proportion of papers funded by NIH that are retracted*.

*This might be a challenging calculation to perform, and maybe the topic of another study, but I would encourage the authors to discuss the point*.

The reviewer seems to be asking in essence whether retracted papers cost, on average, more than non-retracted papers. We initiated a case-control analysis comparing retracted papers to control papers but encountered difficulty defining an unbiased control group. We found that grants that supported retracted papers were generally shorter in duration than those that supported control papers, which biased our sample. A paragraph has been added in the revised manuscript indicating that such a comparison would be of interest and suggest possible reasons why retracted papers may not cost the same as un-retracted controls (Discussion section).

*# “As our estimates of attributable costs are a small percentage of the NIH budget, we identify several reasons why our calculation may underestimate the true financial cost of research misconduct”: the authors do, overall, a thorough job in discussing their results, and show academic rigour in reaching a conclusion that was presumably against their initial hypothesis. Within this context, I find the tone of this sentence a bit out of place. It suggests that the authors, disappointed with the modest costs they had calculated, went on to seek any reason to believe that they underestimated the true costs. In reality, there are reasons (which I point out above) why the costs could have been overestimated as well. Upon revision, more reasons why costs could have been over-estimated could be discussed*.

We have modified this sentence to clarify that our unexpected finding is not attributable to bias. This paragraph reinforces that even after overcorrection for potential bias, we find that research misconduct accounts for a very small percentage of NIH expenditures. In addition, as the reviewer suggests, we have added a paragraph to discuss why our analysis might also have overestimated costs (Discussion section).

*“If only 1% of research misconduct is actually detected … suggesting that research misconduct does not involve a large percentage of research funding in the United States”: This passage is somewhat laboured and would benefit from being shortened*.

This section has been shortened.

“…$1.7 billion in fraudulent expenditures were recovered by Medicaid fraud control units…”: The authors' inclusion, exclusion logic is not apparent. Why not include Medicare fraud? Why mix military misappropriations with research fraud? We suggest deleting this passage

This section has been deleted.

“This indirect financial cost due to the reverberations of fraud throughout the research community is likely to be substantially greater than the cost of the fraudulent research itself”: too speculative, not supported by evidence. I agree that indirect costs of say grants awarded to other scientists based on papers retracted due to misconduct by primary offenders might be high. But since this has not been analyzed, it may be more prudent to say “…throughout the research community 'might be even greater' and not 'is likely to be substantially greater' than the cost…”

We have made the suggested changes.